# The Promises of Speeding Up: Changes in Requirements for Animal Studies and Alternatives during COVID-19 Vaccine Approval–A Case Study

**DOI:** 10.3390/ani12131735

**Published:** 2022-07-05

**Authors:** Merel Ritskes-Hoitinga, Yari Barella, Tineke Kleinhout-Vliek

**Affiliations:** 1Department of Population Health Sciences, Institute for Risk Assessment Sciences (IRAS), Faculty of Veterinary Medicine, Utrecht University, Postbus 80163, 3508 TD Utrecht, The Netherlands; 2Department of Clinical Medicine, Faculty of Health Sciences, Aarhus University Hospital, Palle Juul Jensens Boulevard 99, 8200 Aarhus N, Denmark; 3Faculty of Science, Radboud University, Postbus 9010, 6500 GL Nijmegen, The Netherlands; yaribarella.94@gmail.com; 4Copernicus Institute of Sustainable Development, Utrecht University, Postbus 80.115, 3508 TC Utrecht, The Netherlands; t.h.kleinhout-vliek@uu.nl

**Keywords:** COVID-19, mRNA, vaccines, safety, efficacy, regulations, animal studies, alternatives

## Abstract

**Simple Summary:**

The COVID-19 pandemic led to intensive research into finding new vaccines for human protection. On 21 December 2020, the European Commission granted conditional marketing authorisation for the messenger RNA vaccine ‘Comirnaty’, produced by Pfizer (New York, NY, USA) and BioNTech (Mainz, Germany). This happened only twelve months after the first identification of the virus, whereas the development and approval of vaccines usually take ten years. Through document analysis and interviewing key expert stakeholders, we examined whether the role of animal studies and alternatives in this fast approval process had changed and, if so, whether this could lead to using fewer animal studies and more alternatives in the future. It turned out that in this case, for vaccine development and production, the number of animal studies performed and required had indeed declined, more alternatives had been used and accepted, human studies started earlier and ran in parallel with (rather than sequential to) animal studies, and regulators accepted historical data from earlier vaccine research. The Pfizer/BioNTech vaccine case illustrates the tremendous progress in quickly producing and authorising reliable, safe and effective vaccines, using fewer animal studies and more alternatives. It is time to study the broader implementation of these new procedures on a larger scale to benefit animals and humans.

**Abstract:**

On 21 December 2020, the European Commission granted conditional marketing authorisation for the BNT162b2 COVID-19 vaccine ‘Comirnaty’, produced by Pfizer/BioNTech. This happened only twelve months after scientists first identified SARS-CoV-2. This stands in stark contrast with the usual ten years needed for vaccine development and approval. Many have suggested that the changes in required animal tests have sped up the development of Comirnaty and other vaccine candidates. However, few have provided an overview of the changes made and interviewed stakeholders on the potential of the pandemic’s pressure to achieve a lasting impact. Our research question is: how have stakeholders, including regulatory agencies and pharmaceutical companies, dealt with requirements concerning in vivo animal models in the expedited approval of vaccine candidates such as ‘Comirnaty’? We interviewed key stakeholders at the Dutch national and European levels (*n* = 11 individuals representing five stakeholder groups in eight interviews and two written statements) and analysed relevant publications, policy documents and other grey literature (*n* = 171 documents). Interviewees observed significant changes in regulatory procedures and requirements for the ‘Comirnaty’ vaccine compared to vaccine approval in non-pandemic circumstances. Specifically, the European Medicines Agency (EMA) actively promoted changes by using an accelerated assessment and rolling review procedure for fast conditional marketing authorisation, requiring a reduced number of animal studies and accepting more alternatives, allowing pre-clinical in vivo animal experiments to run in parallel with clinical trials and allowing re-use of historical data from earlier vaccine research. Pharmaceutical companies, in turn, actively anticipated these changes and contributed data from non-animal alternative sources for the development phase. After approval, they could also use in vitro methods only for all batch releases due to the thorough characterisation of the mRNA vaccine. Pharmaceutical companies were optimistic about future change because of societal concerns surrounding the use of animals, adding that, in their view, non-animal alternatives generally obtain faster, better, and cheaper results. Regulators we interviewed were more hesitant to permanently implement these changes as they feared backlash regarding safety issues and uncertainty surrounding adverse effects. Our analysis shows how the EMA shortened its approval timeline in times of crisis by reducing the number of requested animal studies and promoting alternative methods. It also highlights the readiness of pharmaceutical companies to contribute to these changes. More research is warranted to investigate these promising possibilities toward further replacement in science and regulations, contributing to faster vaccine development.

## 1. Introduction

By September 2021, the European Medicines Agency (EMA) had already approved four vaccines for SARS-CoV-2 [1], and the first vaccine by the end of 2020, only twelve months after the first case of SARS-CoV-2 was identified in December 2019 [2,3,4]. This timeframe is unprecedented, as vaccine development from the discovery phase to market approval by regulatory authorities such as the EMA and the United States Food and Drug Administration generally takes ten to fifteen years [2,5]. Of this, two to four years is spent on pre-clinical testing in in vivo animal models, which yields the data requested by regulatory agencies for assessing a vaccine candidate’s safety and immunogenicity (meaning it elicits an immune response, indicative of a protective effect) before making the step to ‘first in human’ clinical trials [6].

The public debate in the early months of the pandemic featured many claims regarding the use of in vivo animal tests, or the suspected lack thereof, in developing several SARS-CoV-2 vaccines. Many expressed worries that the pharmaceutical companies were cutting corners in the process [7,8]. In response, vaccine developers emphasised that they diligently performed animal studies or equivalents before or in parallel with first-in-human trials [9,10,11]. However, it remains unclear what the exact changes in regulatory requirements were and whether and how we can situate these changed activities in the shift towards using fewer animal studies in favour of non-animal alternatives in vaccine development for other diseases (or even medicine development more generally). 

We took the EMA’s December 2020 approval of the BNT162b2 SARS-CoV-2 vaccine ‘Comirnaty’ by Pfizer/BioNTech as our case study to investigate these changes and their potential broader impact. Our specific questions were: how did the developmental process and the regulatory conditional marketing authorisation approval of the SARS-CoV-2 vaccine BNT162b2 differ in required and submitted animal studies compared to the ‘normal’ (i.e., non-pandemic) situation? Moreover, how do stakeholders see the potential for lasting impact based on these changes, and what reasons do they give? We have answered these questions by analysing relevant documents (*n* = 171) and interviewing a selection of key stakeholders directly involved in this process at the Dutch national level (where we are based, interviews with representatives of the following four stakeholder groups: Regulatory Agencies; Medical Sciences; Politics; Society) and at the European level (representatives of the following five stakeholder groups: Regulatory Agencies; Medical Sciences; Politics; Society; Vaccine developer).

## 2. Materials and Methods

We approached our case study, the EMA’s approval (conditional marketing authorisation) of Pfizer/BioNTech’s BNT162b2 SARS-CoV-2 vaccine ‘Comirnaty’, through document analysis and semi-structured interviews.

### 2.1. Document Analysis

As the first step, we performed a document analysis [12]. We gathered as many relevant documents as possible during February–August 2021 by searching Google Scholar, Pubmed, ResearchGate and Google (for media articles, minutes, et cetera) using the search terms described in Table 1. Colleagues in animal research and interviewees supplemented documents to a total of 171. 

The selected documents included guidelines and standards on the development of (COVID-19) vaccines by the World Health Organization (WHO) and EMA, policy documents at the European Union level, minutes from international stakeholder meetings, peer-reviewed articles on vaccine development, COVID-19 vaccines, animal studies (in vaccine development), alternative studies (in vaccine development), the role of stakeholders in determining the role of animal studies and alternatives in vaccine development, and international newspaper articles on the development of COVID-19 vaccines and the role of animal studies and alternatives in their development. 

Based on the documents, we produced an overview of the usual requirements in terms of animal studies in vaccine market authorisation. The document dataset also informed the questions for semi-structured interviews on the current role of animal studies and alternative methods in developing vaccines and their regulatory approval and focused on the Comirnaty vaccine in particular (more information below). 

### 2.2. Semi-Structured Interviews

We approached the following key stakeholder groups for a semi-structured interview: representatives from regulatory agencies at the national and European levels, vaccine developers, (bio)medical scientists/virologists working on animal models or alternative methods, animal rights organisations, and politicians at the national and European level. We identified the key stakeholder groups using a power-interest grid stakeholder analysis. Not all contacted representatives were available for an interview, citing time constraints, but two agreed to reply with a written statement. We conducted two ‘test’ interviews to develop the topic list, refine the questions, and determine which specific stakeholders to approach. We interviewed eleven representatives of five stakeholder groups (Table 2). Under Dutch law, formal ethical review is unnecessary for any research that does not contain a medical-scientific research question and does not expose humans or animals to a behavioural intervention or treatment. When asked, medical ethical review boards in the Netherlands generally consider research proposals like this one ‘exempt from review’. As it concerns a small, explorative interview-based study, the authors did not obtain such an ‘exempt from review’ statement for this study. The interviewees have provided written consent in advance to be interviewed, oral consent to be audio-taped at the time of the interview in line with GDPR art.6.1.a, and were given the opportunity to comment on the near-final version of the paper. 

Our topic list (Appendix A) included the role of animal studies and alternatives for the development and approval of the Comirnaty vaccine compared to vaccine development under ‘normal’ non-pandemic circumstances. Moreover, we asked about the involvement of different stakeholders in (changing) the vaccine development processes and interviewees’ perspectives on lasting impact. 

We transcribed the interviews verbatim and analysed the content of all transcribed interviews and documents through inductive thematic coding in ATLAS.ti [13]. A list of codes describing the themes used can be found in Appendix B. 

## 3. Results

### 3.1. Traditional Vaccine Development and Approval Process

The development of a novel vaccine candidate takes, on average, ten to fifteen years from pathogen discovery to the first authorised vaccination [14]. Regulators generally demand results from animal studies in vaccine development to show vaccine candidates’ efficacy and safety [15].

The vaccine development process starts with an exploratory phase in which basic laboratory research studies obtain information about the pathogen [16]. This knowledge contributes to the laboratory-based development of one or more vaccine candidates. During the pre-clinical phase that follows, selected vaccine candidates are subjected to in vitro, in vivo and in silico studies to show strong evidence of effectiveness, safety, and the ability to elicit an immune response (immunogenicity) before moving into first-in-human trials (Phase 1 trials). This phase features studies such as toxicology testing, dose-ranging, and quality control testing in in vivo animal models considered representative of humans [17,18]. In the final phase, the vaccine candidate is tested in clinical trials (Phase 1, 2 and 3) to show safety, immunogenicity, and efficacy in humans [6]. 

To market a vaccine, one needs marketing authorisation. The requirements for complete marketing authorisation for vaccines in the European Union are outlined in Annex 1 of Directive 2001/83/EC [EUR-Lex]; [19]. Vaccines must be approved through EMA’s ‘centralised procedure’: developers apply for market authorisation at the EMA by submitting all data required, including data gained in the exploratory, pre-clinical, and clinical phases [16]. In addition, validation of the vaccines is required at every point in the manufacturing process to guarantee the new batches are equivalent to the vaccine candidate used in the pre-clinical and clinical trials (interviewees 1 and 4, representing medical sciences and society).

### 3.2. Comirnaty Vaccine Development and Approval Process: Changes in Animal Studies and Alternatives

The development process of the Comirnaty vaccine included all stages described above. However, our dataset highlights notable differences regarding the timeline, animal study requirements, and acceptance of alternative methods (see Figure 1 for vaccine development under ‘normal’ circumstances and Figure 2 for vaccine development under ‘pandemic’ circumstances). 

The European Union has two primary procedures for authorising medicines during pandemics: the mock-up and the emergency procedure [16]. The approval of Comirnaty followed the emergency procedure and used provisions under EU legislation for emergencies. These provisions encompass accelerated assessment (taking a maximum of 150 instead of 210 days), a rolling review, and conditional marketing authorisation [21]. The EMA granted conditional marketing authorisation in approximately 70 days by accelerated assessment and rolling review. The accelerated assessment was first combined with a rolling review, implying that data were analysed as soon as they became available rather than offering all data in one go (interviewee 11, regulatory agencies representative) [16]. Then, once the evidence was deemed sufficient, EMA granted a conditional marketing authorisation intended for use in an emergency in response to public health threats duly recognised by the World Health Organisation and the European Union. The conditional market authorisation was conditional upon data on specific aspects of quality, safety, and efficacy becoming available (interview regulatory agency representative). Throughout the process, the substantial funding available for COVID-related research enabled the generation of the necessary data to rapidly proceed to the next phase of research and development (interview virologist/medical science expert). 

Second, our data show that the exploratory phase of SARS-CoV-2 research featured increased use of in vivo human data and in vitro and in silico methods. These data replaced some in vivo animal studies, with EMA requiring only those animal studies considered to be ‘essential’. Interviewee 2, representing medical sciences, stated that for safety reasons, animal studies are still considered ‘essential’, and efficacy testing can often be tested directly in humans, e.g., via dose escalation studies. According to interviewee 6, representing society, the International Coalition of Medicines Regulatory Authorities (ICMRA) stated there is no need for showing efficacy in animals before progressing to clinical trials. Moreover, the EMA allowed human-specific data to substitute for animal data when of sufficient quality. Our dataset highlighted several technical advances supporting this, such as the complete sequencing of the SARS-CoV-2 genome and, especially, in silico method development. Scientists used these latter methods to establish COVID-19 characteristics, e.g., how the virus infects the host [22], to discover [23], test vaccine candidates [24,25], and determine their efficacy for newly emerging mutated viral strains [26]. According to two interviewees, the achieved reduction in animal studies during the exploratory phase implied that in terms of, e.g., efficacy testing, starting to test in humans earlier might be ‘responsible practice’ also in non-pandemic situations (interviewees 4 and 5, representing medical sciences and society). Certain animal studies could be avoided entirely in future (interviewee 2, medical sciences representative).

Third, we observed a difference in the number of animal studies performed during the pre-clinical phase of vaccine development [27]. For the Comirnaty vaccine, we identified four studies in the pre-clinical phase: two studies (R-20-0085 and R-20-0112) involved mice, one (VR-VTR-10671) concerned rhesus macaques, and the fourth study (20-0211) was an in vitro study. A further four non-clinical in vivo studies involved either mice (R-20-0072) or rats (PF-07302048, 38166, and 20GR142), providing data on pharmacokinetics (Medicines and Healthcare products Regulatory Agency UK) [28]. We could identify no toxicokinetic studies performed in the pre-clinical phase, consistent with WHO guidelines on the non-clinical evaluation of vaccines [29]. No genotoxicity and carcinogenicity studies were performed, as all components of the vaccine constructs are lipids and RNA and are, as such, not expected to have genotoxic, carcinogenic, or tumorigenic potential [29]. No separate studies have been performed to determine local tolerance or generate data on prenatal and postnatal development, including maternal function, dosing or further evaluating offspring, which is why, initially, the vaccine was not recommended for pregnant women. Vaccines are generally thought to hardly pass through the placenta, but we need more research into the risks of vaccinations during pregnancy (interviewees 10 and 11, regulatory agency representatives). No data on reproductive toxicity was provided during authorisation either, as it was not deemed necessary at the time of approval. 

According to our interviewees, the process described above is a drastic reduction in the number of animal studies compared to ‘normal’ (interviewees 1, 2, 4, 5, representing medical sciences and society). To explain this reduction in animal studies in the pre-clinical step, interviewees highlighted the role of the thorough characterisation of mRNA vaccines like Comirnaty. In their view, this led to a better understanding of the product and optimised decision-making regarding safety and efficacy measurements (interviewees 1, 4, representing medical sciences and society). All available non-clinical data (animal and in vitro studies) were assessed in the first round of the rolling review procedure. The remainder of the non-clinical studies were performed later, during Phase 3 or even after vaccine approval (interviewee 10, regulatory agency representative) [30,31].

The thorough characterisation of mRNA vaccines like Comirnaty also positively affected the fourth element, batch release testing. In vaccine development, such batch release testing still often relies partly on animal testing due to the relatively poor characterisation of vaccines [32] and relatively poor product consistency (containing mitigated pathogens), requiring in vivo batch safety and batch potency tests (interviewees 1 and 4, representing medical sciences and society). Instead, the Comirnaty vaccine relied on in vitro alternatives for batch release testing (interviewees 1 and 4, medical sciences and society representatives) [33]. Interviewee 4 added: “The COVID-19 vaccines, much like most modern vaccines, are exceptionally well-characterised and therefore rely on the use of in vitro methods for the batch release testing”. These in vitro alternatives are more precise, robust, cheaper, and have a shorter turn-around time than in vivo animal assays [32,34]. They are also potentially more societally (and scientifically) appropriate (interviewees 4 and 9, medical sciences, society and vaccine developer representatives).

A fifth and final difference concerns the re-use of previous data throughout the approval process, which again benefited from the highly characterised nature of mRNA vaccines. The EMA did not consider it reliable or responsible to avoid animal studies entirely because of the need to demonstrate the product’s safety, immunogenicity, and efficacy. However, regulatory agencies accepted the submission of data previously acquired during research on other mRNA vaccines (interviewees 10 and 11, regulatory agency representatives) [30]. Specifically, ICMRA allowed Pfizer/BioNTech (and other companies) to replace required data with data obtained in pre-clinical in vivo animal studies from mRNA vaccines with similar action mechanisms, in this case, one previously developed for rabies (interview regulatory agency representatives) [30,31,35].

## 4. Discussion

Animal studies take much time, which is one reason why developing and approving vaccines are usually lengthy processes. The EMA’s conditional authorisation of the Comirnaty vaccine indicates that–next to regulatory flexibility, sufficient funding and significant international cooperation—using non-animal alternatives throughout the research, development, and registration processes aided in accelerating vaccine development and approval. These alternatives included, a.o., in silico data for the exploratory phase (now free from animal studies) and using previously acquired data in the pre-clinical phase. Moreover, the EMA allowed moving to in-human trials before completing the ‘essential’ animal studies still required. The use of mRNA technology, which one of our interviewees considered the most significant breakthrough of this pandemic, was vital to enabling many of these changes, though, notably, other vaccines using other technologies were developed and approved in similar timeframes. Interviewees also noted that the pandemic pressure led to more funding, knowledge sharing, and cooperation. As the pandemic pressure led to such widespread influence globally, with the involvement and cooperation of a variety of different stakeholders at multiple levels as well as efficient and fast data sharing and medical progress, it would be valuable to do further transition research into these processes in order to learn how we can apply these lessons more broadly [36,37]. COVID-19 was a global pandemic (making available a significant amount of real-world human data), probably facilitating speedy regulatory approval. 

Nevertheless, this raises the critical question of how these ways of reasoning might hold in future non-pandemic situations, using both mRNA and other technologies. Pharmaceutical companies are ready to take the next step: they seem to consider non-animal alternatives better, cheaper, faster, and more societally salient. When asked about hurdles to implementation, interviewees from all stakeholder groups highlighted risk aversion among regulatory bodies as the primary reason for still requiring animal studies (the “belt and braces approach”), considering these necessary to prove that a product is safe and effective. The critical question is whether alternatives cause higher or lower translational risk than animal studies, as some approved vaccines based on animal studies have shown adverse effects on humans (interviewees 10 and 11, regulatory agency representatives, mentioned the example of narcolepsy after vaccinating against the Mexican flue). Many of our interviewees consider continued dialogue with authorities essential for promoting future acceptance of non-animal alternatives to tackle these risk perception issues. We suggest further quantitative and qualitative research to continue and accelerate these promising processes in obtaining new, better vaccines faster using modern technologies to benefit animals and humans. It will also need further research on how these expedited processes for the mRNA Comirnaty vaccine could work for other conventional vaccine technologies or whether the mRNA technology could help to replace these. Open science, giving complete transparency on methodologies and data, is essential for better decision-making.

## 5. Conclusions

The pressure of the COVID-19 pandemic has shown significant acceleration of the approval process of allowing new vaccines to come to market. The Pfizer/BioNTech vaccine Comirnaty was developed and approved in twelve months, whereas under ‘normal’ conditions, vaccine development and approval take about ten years. In our case study on this fast approval process, we performed document analysis and expert interviews, revealing that regulatory agencies requested fewer animal studies and accepted more alternatives. This also included moving to human trials faster (without awaiting results from animal studies) and accepting historical data from earlier vaccine research. As far as we know, until now, the marketed COVID-19 vaccines have shown promising efficacy and safety. These are all auspicious developments necessitating more research on how to continue using fewer animal studies and more alternatives in the developmental chain towards new, faster and better future vaccines for humans.

## Figures and Tables

**Figure 1 animals-12-01735-f001:**
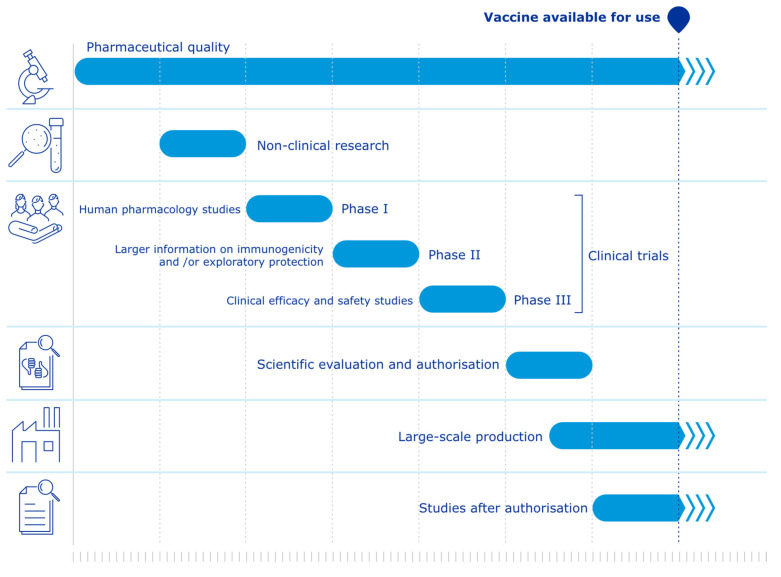
Vaccine development under ‘normal’ circumstances © EMA [1995–2022] [20].

**Figure 2 animals-12-01735-f002:**
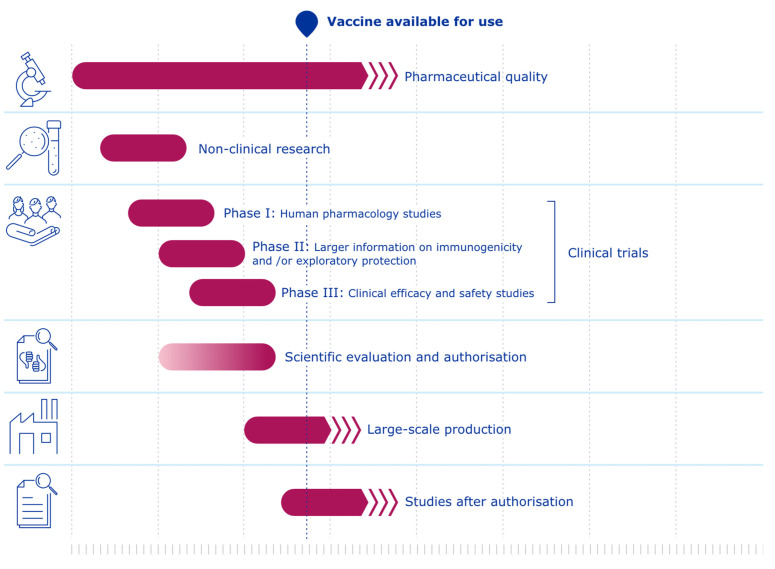
Vaccine development under ‘pandemic’ circumstances © EMA [1995–2022] [20].

**Table 1 animals-12-01735-t001:** Search terms used.

(Animal Testing OR animal studies OR in vivo OR animal experimentation OR animal research)
AND
(COVID-19, COVID, coronavirus, SARS-CoV-2, coronavirus disease 2019, COVID 19, severe acute respiratory syndrome coronavirus 2, BNT162b2, Comirnaty, BioNTech COVID-19 vaccine, Pfizer COVID-19 Vaccine, vaccine, vaccine development, vaccine process, vaccine timeline, vaccine development process, vaccination, nonclinical, non-clinical, pre-clinical, preclinical, nonclinical research, non-clinical research, pre-clinical research, preclinical research)
AND
non-human-animal, non-human-primate, primate, monkey, macaque, rhesus macaque, mice, mouse, rat, rats, ferret, ferrets, hamster, hamsters, rodent, non-rodent

**Table 2 animals-12-01735-t002:** Stakeholder groups interviewed.

Stakeholder Group	Number of Representatives	Type of Data
Medical sciences	7	Interviews
Society	2	Interviews
Politics	2	Interview/written statement
Vaccine developer	1	Written statement
Regulatory Agencies	3	Interviews

## Data Availability

Supporting data have been provided in the appendices. Full interview transcripts (in Dutch) are not provided to guarantee the anonymity of the interviewees, but may be obtained upon reasonable request.

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
