# Peer review of "The Promises of Speeding Up: Changes in Requirements for Animal Studies and Alternatives during COVID-19 Vaccine Approval–A Case Study"

_animals, 2022, doi:10.3390/ani12131735_

Round 1

Reviewer 1 Report

The authors arwe using th Pfizer/BioNTech vaccine case to illustrate how technical progress in molecular genomics allowes to produce and authorise reliable, safe and effective vaccines, using fewer animal studies and more alternatives. They show that It is time to implementat these novel procedures for vaccine production on a larger scale to benefit animals and humans.
Basically they show that advances in basic science allow to use non-animal technologies for the development of vaccines, which the established animal methods will never allow. 
Thus, the authors prove that in the first place the new non-animal methods must be used for scientific and medical reasons and that as a consequnce the use of animals for vaccine production should be abandoned.

Author Response

We are very grateful for the positive comments made by reviewer 1.

Reviewer 2 Report

This is a well written and carefully thought out paper that presents interesting, important and novel data. I congratulate the authors on carrying out this research as it is not easy to engage a wide variety of stakeholders, to get responses and to carry out these analyses but we cannot underestimate the importance of this research and in developing more case studies like this to give an evidence base to the advantages of non-animal approaches.

I have some general comments, as follows, but I have no major issues with the paper and wish you luck in publication and your future work.

I am not sure that I understand the title - I am not familiar with the phrase 'values up' and so I think the authors could refine the title to make it clearer.

The simple summary is very clear and well written.

The abstract could be shortened, I am not sure that there is a need to describe all five steps here.

Abstract line 45: “reducing” does not make sense - suggest rewording to “a reduction in”.

Line 46 (and various other including but not limited to lines 70, 74, 166, 170, 208, 209, 225): in vivo/in vitro/in silico should either all be in italics, or all normal font for consistency.

Line 65 appears to conflict with the abstract that declares marketing authorisation for one vaccine and here it mentions that there are four approved. Could the authors clarify?

Line 71: Could the authors adjust the wording as these data are required by regulators, not necessarily indicative of human immunogenicity and therefore I disagree that animal data are “necessary data for assessing a vaccine candidate’s safety and immunogenicity”

Line 76: “dreading” is a very strong word and I am not sure it is appropriate here

I would find it helpful to have a brief summary table listing the various regulatory agencies and their oversight/jurisdictions.

The actual results of the literature search would be good to see – how many hits, how did the authors refine these? Who read them and who confirmed the analysis?

Figure 1 needs a timeline – even adding years/months would help the authors get the scale of accelerated development across

I found the text at lines 190-196 confusing, I think it might be clearer if the authors first explain the EU procedure and then g on to describe EMA’s use of rolling review – maybe move lines 190-191 to after lines 196?

Line 201: this raises a very important point- could the authors comment further on what was considered essential and how does this differ from non-essential! Why are these data submitted if they are non-essential/not used for risk assessment and how might we change this in the future to save time and animals?

Line 223: I do not understand the significance of using study numbers as quoted– are these used by the regulatory agencies or are the authors referring to study numbers so that the reader may access the same studies?

Line 235-236: “No data on 234 reproductive toxicity was provided during emergency use authorisation. It was not deemed necessary at the time of approval but may have been later.”  I would be interested to know if the authors could comment further on this and do they know if reproductive toxicity studies are ongoing- particularly given the call for vaccine rollout for younger children?

Line 255: Close bracket after ‘Society’

Line 276: The statement that “Animal studies regularly provide delays in vaccine development” is very bold - I think that vaccine developers would disagree with the term “provide delays” and so maybe this could be rephrased.

Line 277: the statement that “The EMA conditional authorisation process of the Comirnaty vaccine shows that using non-animal alternatives throughout the research, development, and registration processes accelerates vaccine development.” is only partly correct as it seems that regulatory flexibility, funding availability and co-operation internationally were also factors here. So although I applaud the authors for pushing the non-animal approaches here, I feel that this statement needs to be tempered for accuracy.  

Lines 283-4: the statement “Interviewees also noted that the pandemic pressure led to more funding, knowledge sharing, and cooperation.” These additional factors are so important and I think they could be explored in more depth. I wonder if the authors could comment further on how we might use the lessons of the pandemic? What do they think could/should alter to enable acceleration beyond vaccines? Do the authors see this regulatory flexibility being used for other vaccines and if not, how can we promote this? For funding, what would the authors recommend or did they get any insight into what the participants would require? Also, given this report https://www.nature.com/articles/d41586-022-01692-1 (which was only published June 21st, I am not expecting the authors to be aware of it of, but would be interested in their thoughts on how this fits what they were hearing in the interviews) - how do we promote data sharing and ensure that it happens? 

I would also be interested in some deeper analysis regarding the fact that COVID-19 was global – meaning that in vivo human data were abundant and maybe this helped regulatory decision making in reducing animal use, given the rich human real world data available whereas the same was not true, for example for Ebola or other non-pandemic viruses. 

Line 56: The authors comment that the regulators fear backlash and I would be interested in exploring this point in more depth. Do the authors know if this is because the regulators themselves do not trust the accelerated process? I think that this analysis could have more impact if the authors could use the discussion points I raise (on funding, data availability, co-operation, etc) to provide a statement on how to change moving onwards as there are implications for future use of non-animal/accelerated approaches that are not fully explored/articulated.

Author Response

Please see the attached file with our responses.

Reviewer 3 Report

The authors are commended for submitting a well-written manuscript regarding a very important but less discussed topic of promising changes in expediated approval of vaccine development. The author summarized and compared 5 main differences between traditional and expediated approval protocols under pandemic situation with very thoughtful analysis and discussions. I believe this is an important and timely analysis and is appropriate for this journal.

Some minor suggestions or comments that may worth some more discussions:

1)    The Generalizability of this expedited vaccine development protocol to other conventional Types of Vaccine Technologies including Inactivated, Live-attenuated, or Subunit, recombinant vaccines remain concerns which may still greatly rely on traditional animal-based model.

2)    Although increased use of in vitro and in silico method in expedited vaccine development benefits for both animal welfare and researchers, current available advances on animal-free alternatives have not progressed enough to be able to replace conventional animal research. Disease like COVID-19 affects several organs and systems, current animal-free alternatives may not come close to this complexity. Weakening/reducing role of animal studies rise more safety concerns.

3)    Importance of transparency on expediated vaccine development protocol especially in animal research.

Author Response

It goes without saying that we really appreciate the positive comments by reviewer 3. We have modified the text according to the suggestions made. Concerning point 2, we agree that alternatives may not come close to complexity. However, animal studies also provide no full guarantee for good predictions for what happens in humans: as we have stated in the text, e.g. Mexican flue vaccine (tested in animals) causes narcolepsia as a side effect in humans. We therefore need evidence-based research for what works and what not.